# A Cross-Sectional Study: Determining Factors of Functional Independence and Quality of Life of Patients One Month after Having Suffered a Stroke

**DOI:** 10.3390/ijerph20020995

**Published:** 2023-01-05

**Authors:** Josefa González-Santos, Paula Rodríguez-Fernández, Rocío Pardo-Hernández, Jerónimo J. González-Bernal, Jessica Fernández-Solana, Mirian Santamaría-Peláez

**Affiliations:** Department of Health Sciences, University of Burgos, 09001 Burgos, Spain

**Keywords:** stroke, quality of life, functional independence, upper limb functionality, determining factors

## Abstract

(1) Background: loss of quality of life (QoL) and functional independence are two of the most common consequences of suffering a stroke. The main objective of this research is to study which factors are the greatest determinants of functional capacity and QoL a month after suffering a stroke so that they can be considered in early interventions. (2) Methods: a cross-sectional study was conducted which sample consisted of 81 people who had previously suffered a stroke. The study population was recruited at the time of discharge from the Neurology Service and Stroke Unit of the hospitals of Burgos and Córdoba, Spain, through a consecutive sampling. Data were collected one month after participants experienced a stroke, and the main study variables were quality of life, measured with the Stroke-Specific Quality of Life Measure (NEWSQOL), and functional independence, measured with the Functional Independence Measure-Functional Assessment Measure (FIM-FAM). (3) Results: the factors associated with a worse QoL and functional capacity one month after having suffered a stroke were living in a different dwelling than the usual flat or house (*p* < 0.05), a worse cognitive capacity (*p* < 0.001) and a worse functional capacity of the affected upper limb (*p* < 0.001). A higher age was related to a worse functional capacity one month after suffering a stroke (*p* = 0.048). (4) Conclusions: the type of dwelling, age, cognitive ability and functional capacity of the affected upper limb are determining aspects in functional independence and QoL during the first weeks after a stroke.

## 1. Introduction

According to the World Health Organization (WHO), a cerebrovascular accident (CVA) or stroke is defined as “a cerebrovascular disease with clinical signs of focal disorders of brain function, that develops rapidly, with symptoms lasting 24 h or more or leading to death, with no other apparent cause than a vascular origin” [1].

This condition continues to be the second leading cause of death and the third leading cause of death and disability combined worldwide. Globally, a cost of more than USD 891 billion (approximately, 1.12% of global GDP) is estimated [2]. In Spain, CVA has been the third highest cause of death since 2020 (with a higher prevalence in the female gender), with COVID-19 and ischemic heart diseases ahead, according to the latest data from the National Institute of Statistics (INE) [3]. According to data from Iberictus, there is an annual incidence of 187 per 100,000 inhabitants [4], although according to other studies based on INE data the annual stroke incidence is 252 per 100,000 inhabitants. Likewise, in Spain it is considered the first cause of disability in adulthood and the second of dementia, seriously impacting the lives of patients and families and causing a significant health and social burden [1,5].

Stroke is a multifactorial disorder associated with a series of risk factors which can be classified into modifiable (such as hypertension, dyslipidemia, lack of physical activity, alcohol consumption, smoking) and non-modifiable (such as age, sex, race, ethnic group or genotype before stroke) risk factors [1,6,7]. It is important to add that, after age, hypertension is the most associated factor. Likewise, obesity should be considered because in addition to being an independent risk factor, it is also a powerful determinant of the evolution of stroke [1].

As expected, stroke has a great impact on both the physical and mental health of the affected patients and their quality of life (QoL), which is directly related to the severity of the episode and the number of comorbidities that the patient suffers. Therefore, the assessment of physical and mental health is of great importance to determine both the consequences and the possible treatment of stroke [8].

According to the International Classification of Functioning, Disability and Health (ICF), stroke can cause impairments in function and alterations of body structures that lead to functional limitations [9]. Significant impairment in upper limb function can be observed in more than 80% of stroke survivors [10,11]; which, in turn, is usually associated with the QoL in all its areas. This acquired deficit can significantly impair Activities of Daily Living (ADL), so that between 23% and 53% of patients have total or partial dependence [11]. These can determine the level of functional impairment that the patients have in their daily lives, and are a clinically relevant outcome measure to assess the level of recovery after stroke [12,13].

Some of the most common physical consequences that can be found in patients who have suffered a stroke are muscle weakness or paralysis in the affected hemibody, impaired muscle coordination, an alteration in superficial, exteroceptive and proprioceptive sensitivity, pain that can limit the joint range of motion (ROM), alterations in the tone of the musculature and/or an apparent deformity of the entire upper limb [11,14]. On the other hand, the possibility that a stroke can cause a certain degree of cognitive impairment has been also confirmed. The prevalence of post-stroke cognitive impairment is high and both demographic (age, education and occupation) and vascular factors represent risk factors for cognitive impairment after stroke [15,16], significantly affecting the QoL of patients and their functional recovery capacity [17,18].

Regarding its prognosis, multiple factors affect the development and functional recovery of the upper limb, such as the great variability in performance between individuals, cognitive, sensory and motor function, psychological factors, age, gender and the dominant hand [19]. Functional impairment is the most common consequence, in addition to cognitive impairment [20], which can lead to greater dependence on ADL performance, job loss and decreased QoL [21]. Therefore, the main objective of this research is to study which factors are related to functional capacity and QoL a month after suffering a stroke, so that these can be considered in early interventions.

## 2. Materials and Methods

### 2.1. Participants

This cross-sectional study was conducted in Burgos and Córdoba (Spain). The sample consisted of 81 people who had previously suffered a stroke. As inclusion criteria, it was required that the participants were older than 18 years and had a diagnosis of residual hemiparesis due to stroke, where movement of the affected upper limbs was classified between stages II and IV of the Brunnstrom Scale [22].

The study population was recruited at discharge from the Neurology Service and Stroke Unit of both hospitals through consecutive sampling.

The sample size was estimated following the procedure for finite populations, using the formula *n* = N × (Zα = 1.96) × 2 × p × qδ2 × (N − 1) + (1.96)^2^ × p × q. The known population reported by the National Institute of Statistics (INE) [23] was taken into account, establishing a proportion of strokes in the population and assuming a sampling error of 1%. Based on this, it was concluded that the sample should be made up of 81patients with strokes under care at the University Hospital of Burgos (HUBU).

### 2.2. Instruments

For the statistical analysis, a series of quantitative and qualitative variables related to social and clinical aspects of interest were used. The main variables were functional independence and quality of life one month after having suffered a stroke.

To assess the degree of independence in ADLs, the Functional Independence Measure-Functional Assessment Measure (FIM-FAM) was used [24,25,26]. This is a scale that assesses the degree of global disability with a Cronbach’s alpha of 0.968 [27]. It contains 18 items that can be used either independently or attached to 12 items belonging to the FAM. It scores on an ordinal scale from 1 (total dependence) to 7 (total independence), with a total of 210 points indicating greater functional independence [24,25,26]. 

Regarding quality of life, the Stroke-Specific Quality of Life Measure (NEWSQOL) was used [28]. This is the first questionnaire translated into the Spanish language to specifically measure QoL in patients with a stroke. It includes 38 items grouped into eight domains (physical state, communication, cognition, emotions, feelings, ADLs, IADLs and socio-family functions) rated from 1 (no difficulty in performing tasks) to 5 (extreme difficulty in performing tasks), so that a higher score will be indicative of worse results in the QoL of the person. The NEWSQOL has excellent values of internal consistency, with Cronbach’s alpha coefficients between 0.88 and 0.97 [29]. This Spanish version has demonstrated good validity and reliability [28].

Regarding the independent variables, the Fugl-Meyer Assessment of the Upper Extremity (FMA-EU) was used to assess functional capacity [30,31,32]. It is a scale translated into Spanish and validated in the Spanish population, with good reliability and validity and a Cronbach’s alpha of 0.973. It consists of 33 items distributed in four domains (motor, sensory, range of motion and pain), which score from 0 (non-realization) to 2 (complete realization), with a total score of 66 points and partial scores of 36 points for the proximal part of the arm and 30 for wrist and hand. Higher total scores on the scale mean increased upper limb functionality, normal exteroceptive and proprioceptive sensitivity, a correct range of passive mobility and no pain [30,31,32].

For the cognitive assessment, the Montreal Cognitive Assessment (MOCA) was used [31,32,33,34]. It evaluates six cognitive domains (memory, visuospatial function, executive function, attention/concentration/working memory, language and orientation). Its maximum score is set at 30 points, the maximum score being the non-existence of cognitive impairment (Cronbach’s alpha of 0.70). It is always necessary to take into account when using this evaluation the educational level of the person and the cut-off score should be placed at 26 to assess the existence of cognitive impairment [35,36].

Sociodemographic data, such as age, sex, nationality, level of education, marital status, number of children, place of residence, type of dwelling, with whom a participant lives, tobacco use, alcohol consumption or physical activity; general clinical variables, such as hypertension, dyslipidemia, myocardial infarction, diabetes mellitus (DM) or obesity; and clinical variables of interest after stroke such as the dominant and affected side and cognitive ability and functionality of the upper limbs at one month after stroke were obtained.

### 2.3. Procedure

This was a multicenter study performed at the University of Burgos (UBU) in collaboration with the HUBU, Hospital San Juan de Dios (Burgos) and the Reina Sofía Hospital in Córdoba. After signing the collaboration and confidentiality agreement document with the participating centers, the data collection necessary for this research began. The participating centers positively valued the research plan in the IR Approval Committee HUBU 2134/2019. The data obtained were sent to the research team after an anonymization process; from this moment on, they are always treated anonymously and together.

The data were collected just one month after suffering a stroke. All participants signed an informed consent form before starting.

Once the data was obtained, a matrix was created for evaluation using the statistical program Software IBM SPSS (Statistical Package for the Social Sciences) in its 25th version.

### 2.4. Data Analysis

Descriptive analyses of the characteristics of the sample were performed, expressing the categorical variables in absolute frequencies and percentages and the continuous ones in means and standard deviations (Sd). The normality of the dataset was contrasted using the Kolmogorov–Smirnov test. To assess the association between functional independence and quality of life (the main variables of the study) and the categorical variables of two groups, the Student’s T test was used for the independent samples. The association between functional independence and quality of life and the categorical variables of three or more groups was analyzed using the ANOVA test. Pearson’s correlation was used to assess the relationship between the study’s primary variables and the different continuous variables.

Statistical analysis was performed with SPSS software version 25 (IBM Inc., Chicago, IL, USA). For the analysis of statistical significance, a *p*-value < 0.05 was established.

## 3. Results

Data were obtained from a total of 81 patients a month after suffering a stroke, whose mean age was 68.42 ± 12.43 years old. The distribution by sex was homogeneous, with 54.3% men (*n* = 44) and 45.7% women (*n* = 37).

Most of the participants were right-handed (*n* = 78, 96.3%), but 51.9% (*n* = 42) suffered the injury in the right hemisphere, so their affected side was the left one; and the remaining 48.1% (*n* = 39) suffered the injury in the left hemisphere, so their affected side was the right one.

Regarding sociodemographic variables, more than half of the sample had completed basic studies (*n* = 45, 55.6%) and 23.5% (*n* = 19) had no studies. 55.6% (*n* = 46) of the participants were married or were part of a couple at the time of the stroke, while 24.7% (*n* = 20) were single and the remaining 18.5% (*n* = 15) were separated, divorced or widowed. Half of the respondents (*n* = 41, 50.6%) had children. 

Regarding the participants’ place of residence, 81.5% of the subjects resided in Burgos (53 in the city and 13 in towns in the province) and the remaining 18.5% in Córdoba (12 in the city and 3 in towns in the province). The majority lived accompanied (*n* = 75, 92.6%) and 67.9% (*n* = 55) of the total lived in a flat, 22.2% (*n* = 18) in a house, 2.5% (*n* = 2) in a nursing home and 7.4% (n=6) in another type of dwelling such as in a community of religious men or women.

Regarding the clinical characteristics of the study, Table 1 shows the average score obtained in the different quantitative variables. The most frequent comorbidities were arterial hypertension (*n* = 34, 42%), dyslipidemia (*n* = 16, 19.75%), diabetes mellitus (*n* = 16, 19.75%) and obesity (*n* = 16, 19.75%). Only 23.5% (*n* = 19) of the participants practiced physical exercise before the stroke, and 22.2% (*n* = 18) and 12.3% (*n* = 10) were habitual consumers of tobacco and alcohol, respectively.

### 3.1. Functional Independence 

Table 2 summarizes the differences between the independent study variables of two groups in functional independence. No statistically significant differences were found in functional independence depending on the sociodemographic and clinical variables of the two groups (sex, children, with whom they live, practice of physical activity, tobacco and/or alcohol consumption, hypertension, dyslipidemia, atrial fibrillation, myocardial infarction, diabetes mellitus, obesity, dominant side and affected side).

Table 3 summarizes the differences between the independent study variables of three or more groups in functional independence. No statistically significant differences were found in functional independence depending on the level of education, marital status and place of residence, but they were found to depend on the type of dwelling. The average score of participants living in a flat was 166.73 ± 37.80, those living in a house reported an average score of 161.89.36 ± 36.22, those living in a nursing home a score of 118.50 ± 13.43 and those living in other types of dwelling such as religious communities scored an average of 93.83 ± 73.65. An ANOVA test revealed significant differences between those who lived in another type of dwelling with respect to those who lived in a flat or house (*p* < 0.05), the former being the most dependent. The post hoc test indicates significant differences between those who live on the flatand others (*p* < 0.001) and also between those who live at home and others (*p* = 0.002).

Table 4 summarizes the relationship between the different quantitative study variables and functional independence. A significant relationship was found between functional independence and all study variables. Functional independence was negatively correlated with age (*p* = 0.048) and positively correlated with the functional capacity of the affected upper limb (*p* < 0.001); with the subscales of mobility (*p* < 0.001), sensitivity (*p* < 0.001) and range/pain (*p* < 0.001); and cognitive ability (*p* < 0.001).

### 3.2. Quality of Life

Table 5 summarizes the differences between the independent study variables of two groups in quality of life. No statistically significant differences were found in the quality of life depending on the sociodemographic and clinical variables of these two groups (sex, children, with whom they live, practice of physical activity, tobacco and/or alcohol consumption, hypertension, dyslipidemia, atrial fibrillation, myocardial infarction, diabetes mellitus, obesity, dominant side and affected side).

Table 6 summarizes the differences between the independent study variables of three or more groups in quality of life. No statistically significant differences were found in quality of life depending on the level of education, marital status and place of residence, but were found to depend on the type of dwelling. The average score of participants living in a flat was 88.85 ± 30.96, those living in a house reported an average score of 98.00 ± 32.79, those living in a nursing home a score of 126.50 ± 0.70 and those living in other types of dwelling such as in religious communities scored an average of 125.00 ± 37.67. An ANOVA test revealed significant differences between those who lived in another type of dwelling with respect to those who lived in a flat (*p* < 0.05), with the former reporting a worse quality of life. The post hoc test indicates significant differences between those who live on the flatand others (*p* < 0.051).

Table 7 summarizes the relationship between the different quantitative study variables and quality of life. A significant negative relationship was found between quality of life and all quantitative study variables except age. That is, the results showed a higher quality of life in patients with better functionality (*p* < 0.001), mobility (*p* < 0.001), sensitivity (*p* < 0.001), range/pain (*p* < 0.001) of the affected upper limb and better cognitive abilities (*p* < 0.001).

## 4. Discussion 

The main objective of this research was to study which factors are related to functional capacity and QoL a month after having suffered a stroke, so that they can be considered in early interventions.

The high incidence and the burden of stroke disease indicate the importance of urgent control of risk factors for the prevention of stroke [37]. However, other very important points for the prediction of the prognosis in the recovery of patients are the determinants of functional capacity and QoL. Major risk factors include physical inactivity, hypertension, smoking, dyslipidemia and obesity [37,38,39], which may clarify the results found in our study: 42% of the sample had hypertension, 76.5% of them did not perform physical exercise and 12.3% were habitual tobacco users.

With regard to obesity, which affected 19% of the patients in our sample, the studies have focused more specifically on the inverse relationship between BMI and mortality after suffering a stroke. However, in the case of stroke survivors, the association between BMI and functional recovery was unclear [40]. Some studies showed that a BMI above 35 kg/m^2^ was a protective factor of patient independence [41], although, in contrast to our results, Razinia et al. [42] found no relationship between BMI and functional recovery after discharge. Likewise, risk factors are also associated in some way with different QoL scores of patients. However, dyslipidemia does not seem to have a significant effect, perhaps because of its high prevalence in our country [43].

Our results suggest the existence of statistically significant differences between the independence of the patient, measured with the FIM scale, and the type of dwelling in which the patient usually resides. In this case, it can be observed how patients who live in an apartment, compared to other types of dwelling, are more independent in their daily lives. In turn, statistically significant differences have also been observed when comparing the type of habitual dwelling with patients’ QoL (ECVI-38), showing that patients with the best quality of life reside in a flat.

With respect to the level of independence or functional capacity of patients, statistically significant differences have also been found in relation to the functionality of the upper limbs, in all subscales of the FMA-UE evaluation instrument (mobility, sensitivity, amplitude and pain); and cognitive ability, assessed by MOCA. Previous studies report that the degree of motor impairment at baseline may influence the incidence of post-stroke medical complications, which may imply worse functional outcomes months after stroke [44]. Likewise, it is also suggested that other complications, such as the presence of depression, cognitive impairment, aphasia or unilateral negligence, were also factors that affected the recovery of motor function [19]. In other investigations, it can be confirmed that a greater functional impairment and the presence of pre-stroke dependency may also be associated with a worse short- and long-term subsequent functional prognosis [45].

There are mechanisms such as changes in metabolism and cerebral blood flow that may explain the relationship between the level of motor and cognitive activity and the degree of functional independence after a stroke. Studies on the pathophysiology of the brain and the evolution of functional deficits hypothesize that alteration at the level of cerebral blood flow can be used as an inducer of brain reorganization, although the mechanisms underlying neurological recovery are unclear. It may be possible that a better functional state before the stroke could lead to an increase in blood flow and a decrease in damage once it has happened; which, in turn, causes improvements in the general functional status of the patient due to the optimization of brain cells and the reorganization of neuronal activity [46,47].

In reference to the level of independence, a significant negative correlation is also shown with the age variable, where an increase in age determines a worse general functional capacity and a lower level of independence. Poggesi et al. [48] found in their results that age may also be an influential factor in the patient’s functional outcomes after a stroke, although it appeared to be associated with sex, so that men’s functional recovery decreased with increasing age while older women continued to improve regardless of the initial deficit. Previous studies also found that age, gender and type of stroke predict long-term functional outcome after discharge [21,38]. Another study also corroborates our results, indicating that age influences performance and, therefore, may be a prognostic factor for motor function after a stroke [19].

On the other hand, in the analysis of the QoL of the patients in our study, significant results are also obtained in relation to the functionality of the upper limbs in all the subscales of the FMA-EU and the score in MOCA, that is, in the level of cognitive capacity. Better scores in both evaluations are positively related to better scores in the perception of QoL in post-stroke patients. It should be noted that there is no widely accepted consensus on the factors that affect or determine the QoL of stroke patients, but it can be said that stroke significantly compromises the QoL of these patients [43]. Despite this, the results of another study corroborate the data obtained in our research, showing that the QoL of patients can depend significantly on several factors such as the patient’s level of functionality or the MOCA score [49] and that these, according to our results, can be established as risk factors or determinants for QoL and the level of functional dependence in stroke patients [50].

Previous studies indicate that QoL is worse 3 months after the occurrence of a stroke and that the patient’s pre-stroke level is not reached [43,51]. In the same way, there is evidence in other studies of a significant reduction in QoL during the first month after a stroke episode, although it has been seen that certain improvements are subsequently achieved after the third month [52]. As a result, it can be confirmed that this condition has a significant and prolonged disabling effect on QoL, despite improvements in treatment in the acute phase [43].

Various investigations performed in surviving patients associate cognitive impairment with a greater affectation of their QoL. This association is found in more than half of patients, without showing significant variation over time towards the recovery of cognitive status, but rather showing that it worsens [53]. More than half of these patients present an alteration in their QoL between moderate and severe; however, the QOL has not shown a significant relationship with variables such as sex, not having a partner, level of education, economic situation, location of the injury or having comorbidities. However, greater neurological involvement, the presence of cognitive impairment and increased age were influential factors in QoL [53]. According to this study, no significant differences in QoL scores are found in terms of sex [43], although others do affirm the existence of worse QoL scores in women up to 12 months later [54,55].

Therefore, the functional status presented by the patient is usually directly linked to the neurological involvement derived from the injury, and is linked in turn to the involvement of QoL [53]. Consequently, it can be seen how the functional and cognitive state of the patient gives rise to a better perception of QoL and level of independence, and the importance of the control of risk factors can be highlighted to improve QoL globally [43].

As for limitations of our study, we find, on the one hand, that the collection of the sample was performed by means of a non-probabilistic sampling; and, on the other hand, the number of subjects in some of the variables studied was small and, therefore, significant differences may not have been obtained. The need to continue researching in this line is observed, since few studies have been conducted on this topic, and specifically in such early stages of stroke. Taking into account the predictor variables of a better functional capacity and QoL can be a turning point in the lives of patients and, therefore, it would be important to collect more information by expanding the study samples. As for strengths, we should highlight the great variety of variables that have been collected in our sample of patients a month after suffering a stroke, which has allowed us to establish more appropriate intervention objectives in the early stages of stroke recovery for greater independence and QoL. Some sociodemographic variables can be a confounding factor for clinical variables.

## 5. Conclusions

The factors related to a worse QoL and functional capacity a month after having suffered a stroke were living in a different dwelling than the usual flat or house, cognitive impairment and having a worse functional capacity in the affected upper limb, and more specifically a worse sensitivity and mobility, a lower range and greater pain in the affected upper extremity. Increased age was associated with worse functional capacity one month after a stroke, but not with QoL.

That is, the type of home, age, cognitive ability and functional capacity of the affected upper limb are important aspects in functional independence and QoL during the first weeks after having suffered a stroke.

And, therefore, it is important to highlight these data at the clinical level, since it can allow professionals dedicated to rehabilitation to initially evaluate these variables more accurately in order to establish the most appropriate individualized treatment plan adapted to the characteristics that a patient presents, always keeping in mind the impact the approach to these variables can have on the autonomy of a patient’s daily life and QoL.

## Figures and Tables

**Table 1 ijerph-20-00995-t001:** Descriptive data of the evaluation instruments related to the clinical variables of study.

Quantitative Variables	Mean ± Sd
FIM-FAM	159.06 ± 44.57
NEWSQOL	94.49 ± 32.93
MOCA	20.22 ± 7.37
FMA-EU Mobility	44.12 ± 20.58
FMA-EU Sensitivity	6.60 ± 3.46
FMA-EU Range/Pain	42.96 ± 7.75
Total FMA-EU	96.69 ± 26.84

Sd: Standard Deviation; FIM-FAM: Functional Independence Measure-Functional Assessment Measure; NEWSQOL: Stroke-Specific Quality of Life Measure; MOCA: Montreal Cognitive Assessment; FMA-EU: Fugl-Meyer Assessment of the Upper Extremity.

**Table 2 ijerph-20-00995-t002:** Differences between independent study variables of two groups in functional independence using student’s *t* test for independent samples.

Categorical Variables from Two Groups		FIM-FAM			
		Mean	Sd	*t*	*p* Value (Bilateral)
Gender	Female	162.70	41.50	–0.672	0.504
	Male	156.00	47.26		
Children	No	161.03	48.10	0.389	0.698
	Yes	157.15	41.35		
Whom subject lives with	Alone	180.33	30.82	1.219	0.227
	Accompanied	157.36	45.21		
Dominant side	Left	170.67	37.43	0.457	0.649
	Right	158.62	44.67		
Affected side	Left	157.38	49.35	–0.350	0.727
	Right	160.87	39.34		
Physical activity	No	148.76	53.41	–1.882	0.065
	Yes	169.11	30.47		
Tobacco use	No	159.64	50.53	0.370	0.713
	Yes	154.72	40.25		
Alcohol consumption	No	156.44	48.47	-0.603	0.548
	Yes	166.10	34.44		
HTA	No	145.54	55.28	–1.482	0.144
	Yes	163.44	39.66		
Dyslipidemia	No	155.38	51.80	–0.161	0.872
	Yes	157.45	36.74		
Myocardial infarction	No	158.12	47.12	0.715	0.477
	Yes	134.00	38.18		
DM	No	156.20	45.94	–0.180	0.858
	Yes	158.63	51.24		
Obesity	No	155.43	45.21	–0.862	0.392
	Yes	166.81	50.34		

Sd: Standard Deviation; FIM-FAM: Functional Independence Measure-Functional Assessment Measure; HTA: arterial hypertension; DM: Diabetes Mellitus.

**Table 3 ijerph-20-00995-t003:** Differences between independent study variables of three or more groups in functional independence using ANOVA.

Categorical Variables from Three or More Groups		FIM-FAM			
		Mean	Sd	F	*p* Value
Level of studies	No studies	143.58	52.66	1.100	0.354
	Basic	162.49	43.28		
	Upper	171.22	33.85		
	University	162.88	39.51		
Marital status	Bachelor	142.90	56.61	1.858	0.144
	Married/coupled	167.67	36.73		
	Widower	144.13	55.65		
	Separated/divorced	165.71	28.65		
Place of residence	Burgos city	161.00	48.18	0.314	0.815
	Burgos province	155.15	41.68		
	Cordoba city	160.33	33.88		
	Cordoba province	136.67	37.65		
Type of address	Flat	166.73	37.80	6.520	0.001
	House	161.89	36.22		
	Residence	118.50	13.44		
	Other	93.83	73.65		

Sd: Standard Deviation; FIM-FAM: Functional Independence Measure-Functional Assessment Measure. *p* < 0.05.

**Table 4 ijerph-20-00995-t004:** Relationship between the different quantitative study variables and functional independence.

Quantitative Variables	FIM-FAM	
	*r*	*p* Value
Age	–0.225	0.048
Total FMA-EU	0.677	<0.001
FMA-EU Mobility	0.663	<0.001
FMA-EU Sensitivity	0.408	<0.001
FMA-EU Range/Pain	0.407	<0.001
MOCA	0.576	<0.001

FIM-FAM: Functional Independence Measure-Functional Assessment Measure; FMA-EU: Fugl-Meyer Assessment of the Upper Extremity.; MOCA: Montreal Cognitive Assessment.

**Table 5 ijerph-20-00995-t005:** Differences between independent study variables of two groups in quality of life using student’s *t* for independent samples.

Categorical Variables from Two Groups		NEWSQOL			
		M	Sd	*t*	*p* Value (Bilateral)
Gender	Female	95.62	34.39	–0.281	0.779
	Male	93.55	32.02		
Children	No	89.28	32.57	–1.418	0.160
	Yes	99.59	32.87		
Whom subjects live with	Alone	79.50	30.24	–1.162	0.249
	Accompanied	95.69	33.03		
Dominant side	Left	96.33	27.75	0.098	0.922
	Right	94.42	33.26		
Affected side	Left	94.14	33.35	–0.099	0.921
	Right	94.87	32.90		
Aphysical activity	No	97.52	36.72	0.283	0.778
	Yes	94.79	30.66		
Tobacco use	No	89.89	34.37	–1.887	0.064
	Yes	107.78	33.71		
Alcohol consumption	No	95.00	34.13	–0.236	0.814
	Yes	97.80	38.01		
HTA	No	102.39	35.36	1.022	0.311
	Yes	93.29	93.29		
Dyslipidemia	No	94.62	34.22	–0.864	0.391
	Yes	102.70	36.05		
Myocardial infarction	No	95.05	34.74	–0.903	0.370
	Yes	117.50	27.58		
DM	No	98.33	36.23	0.842	0.403
	Yes	90.00	28.20		
Obesity	No	99.94	33.49	2.134	0.036
	Yes	79.69	32.83		

Sd: Standard Deviation; NEWSQOL: Stroke-Specific Quality of Life Measure; HTA: arterial hypertension; DM: Diabetes Mellitus.

**Table 6 ijerph-20-00995-t006:** Differences between independent study variables of three or more groups in quality of life using ANOVA.

Categorical Variables from Three or More Groups		NEWSQOL			
		Mean	Sd	F	*p* Value
Level of studies	No studies	105.00	37.85	0.914	0.438
	Basic	92.36	31.48		
	Upper	86.78	26.59		
	University	90.25	35.26		
Marital status	Bachelor	104.75	30.68	1.893	0.138
	Married/coupled	87.02	31.87		
	Widower	103.43	38.43		
	Separated/divorced	104.43	33.81		
Place of residence	Burgos city	91.00	33.05	1.015	0.391
	Burgos province	102.31	36.32		
	Cordoba city	95.25	25.65		
	Cordoba province	119.33	42.19		
Type of address	Flat	88.85	30.96	3.195	0.028
	House	98.00	32.79		
	Residence	126.50	0.71		
	Other	125.00	37.67		

Sd: Standard Deviation; NEWSQOL: Stroke-Specific Quality of Life Measure; *p* < 0.05.

**Table 7 ijerph-20-00995-t007:** Relationship between the different quantitative study variables and quality of life.

Quantitative Variables	NEWSQOL	
	*r*	*p* Value
Age	0.160	0.162
Total FMA-EU	–0.700	<0.001
FMA-EU Mobility	–0.658	<0.001
FMA-EU Sensitivity	–0.490	<0.001
FMA-EU Range/Pain	–0.462	<0.001
MOCHA	–0.498	<0.001

NEWSQOL: Stroke-Specific Quality of Life Measure; MOCA: Montreal Cognitive Assessment; FMA-EU: Fugl-Meyer Assessment of the Upper Extremity.

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
