# Peer review of "A Cross-Sectional Study: Determining Factors of Functional Independence and Quality of Life of Patients One Month after Having Suffered a Stroke"

_ijerph, 2023, doi:10.3390/ijerph20020995_

Round 1
Reviewer 1 Report
The authors produced an interesting work about factors of functional independence and quality of life of patients one month after a stroke. Despite the small sample size as stated on limitation section by the author they analysed a lot of different variable. This is surely a strenght of the manuscript which could interest the health professionals.
Comments for the authors
-) ‘’The sample consisted of 81 people who had previously suffered a stroke’’ (line 88)
This sentence is related to the result. In this section you have to specify inclusion/exclusion criteria, where do you select the group and if you stated or not a sample size and why do you state it. Improve a little bit this section.
-) On line 144 of the procedure you stated ‘’A randomized controlled….’’ why do you randomize the patient? this is a cross-sectional study so you do not need it. Explain
-) on table one the MIN/MAX value is not necessary
-) on table 2 could be interesting to add the numerosity for each subsample that you analysed.
Author Response
Mrs. Jessica Fernández Solana
Department of health sciences
University of Burgos, Paseo Comendadores s/n.
Burgos, 09001, Spain
Tel. (+34) 947499108
Email: jfsolana@ubu.es
16-12-2022
IJERPH. Subject: Submissions Needing Revision
Dear editor.
Thank you very much for inviting us to submit our response to reviewers for our manuscript (ijerph-2010935) entitled: “A cross-sectional study: determining factors of functional independence and quality of life of patients one month after having suffered a stroke.”
We have checked our manuscript according to the Academic Editor, the reviewers’ comments and the Journal requirements. We have also responded to some comments from reviewers point by point).
We would be very grateful if you could consider our manuscript to be published in your journal.
Yours sincerely,
Jessica Fernández Solana, OT, PT
- Response to Reviewer 1:
First of all, we would like to express our sincere gratitude for all comments and suggestions received from the Reviewer 1. This information has certainly enriched the text for its best understanding, thank you very much indeed. We have clarified the reviewer1’s questions. We have introduced the required changes both in our answers to the specific comments and in the final manuscript V2.
The authors produced an interesting work about factors of functional independence and quality of life of patients one month after a stroke. Despite the small sample size as stated on limitation section by the author they analysed a lot of different variable. This is surely a strenght of the manuscript which could interest the health professionals.
Comments for the authors
-) ‘’The sample consisted of 81 people who had previously suffered a stroke’’ (line 88)
This sentence is related to the result. In this section you have to specify inclusion/exclusion criteria, where do you select the group and if you stated or not a sample size and why do you state it. Improve a little bit this section.
Response: Thank you for your comment. Inclusion and exclusion criteria have been included.
“This cross-sectional study was conducted in Burgos and Córdoba (Spain). The sample consisted of 81 people who had previously suffered a stroke. As inclusion criteria, it is contemplated that the participants are older than 18 years, with a diagnosis of residual hemiparesis due to stroke, those whose movement of the affected upper limbs is classified between stages II and IV of the Brunnstrom Scale [21].
The study population was recruited at discharge from the Neurology Service and Stroke Unit of both hospitals through consecutive sampling.”
-) On line 144 of the procedure you stated ‘’A randomized controlled….’’ why do you randomize the patient? this is a cross-sectional study so you do not need it. Explain
Response: Randomised control was performed for subsequent phases of the study. This information has been removed to avoid confusion in this manuscript.
-) on table one the MIN/MAX value is not necessary
Response: Se han eliminado las columnas en las que se recogían estos valores.
-) on table 2 could be interesting to add the numerosity for each subsample that you analysed.
Response: This information is given as descriptive information in the first part of the results.
We hope we have now answered all your comments and we are looking forward to hearing from you again.
Thank you very much,
Jessica Fernández Solana, OT, PT

Reviewer 2 Report
Thank you for the opportunity to review this interesting paper. Evaluating factors that most influence lives of people with stroke is certainly an interesting topic that can offer insights for health care and rehabilitation.
Intro: I appreciated global data and specific focus on Spain context, also referring to ICF.
Methods:
INstruments: Please divide paragraph in quantiative and qualitative data. First describe qualitative apporach, then quantiative approach and please separate instruments sub-section dividing each outcome measure. This can help readers to follow your manuscript.
Procedures: who administered the outcome measures? Why you need a blind randomized sample in a cross-sectional study? Please clarify
Data analysis: considering you analyzed different variables, why you choose to perform only anova analysis, not also MANOVA? This can help to better understand how these variables can affect QoL in people with stroke.
Author Response
Mrs. Jessica Fernández Solana
Department of health sciences
University of Burgos, Paseo Comendadores s/n.
Burgos, 09001, Spain
Tel. (+34) 947499108
Email: jfsolana@ubu.es
16-12-2022
IJERPH. Subject: Submissions Needing Revision
Dear editor.
Thank you very much for inviting us to submit our response to reviewers for our manuscript (ijerph-2010935) entitled: “A cross-sectional study: determining factors of functional in-dependence and quality of life of patients one month after having suffered a stroke.”
We have checked our manuscript according to the Academic Editor, the reviewers’ comments and the Journal requirements. We have also responded to some comments from reviewers point by point).
We would be very grateful if you could consider our manuscript to be published in your journal.
Yours sincerely,
Jessica Fernández Solana, OT, PT
- Response to Reviewer 2:
First of all, we would like to express our sincere gratitude for all comments and suggestions received from the Reviewer 2. This information has certainly enriched the text for its best understanding, thank you very much indeed. We have clarified the reviewer2’s questions. We have introduced the required changes both in our answers to the specific comments and in the final manuscript V2.
Thank you for the opportunity to review this interesting paper. Evaluating factors that most influence lives of people with stroke is certainly an interesting topic that can offer insights for health care and rehabilitation.
Intro: I appreciated global data and specific focus on Spain context, also referring to ICF.
Response: Thank you
Methods:
INstruments: Please divide paragraph in quantiative and qualitative data. First describe qualitative apporach, then quantiative approach and please separate instruments sub-section dividing each outcome measure. This can help readers to follow your manuscript.
Response: The sociodemographic variables have been moved to the end of the section for a better structuring of the section.
Procedures: who administered the outcome measures? Why you need a blind randomized sample in a cross-sectional study? Please clarify
Response: Randomised control was performed for subsequent phases of the study. This information has been removed to avoid confusion in this manuscript.
Data analysis: considering you analyzed different variables, why you choose to perform only anova analysis, not also MANOVA? This can help to better understand how these variables can affect QoL in people with stroke.
Response: Thank you for your comment. As this was a cross-sectional study, it was considered more appropriate to use an ANOVA test rather than MANOVA.
We hope we have now answered all your comments and we are looking forward to hearing from you again.
Thank you very much,
Jessica Fernández Solana, OT, PT

Reviewer 3 Report
Dear Authors,
Thank you for the opportunity to revise your manuscript. While the topic is relevant, there are serious concerns about the rational, study design, statistics and conclusions.
Please find below major issues:
- In a cross-sectional study is not possible to draw casual relations between variables and outcomes. It is a correlation, that does not mean that a variable causes a change in the outcome. A controlled prospective study is needed.
- How the variables were selected?
- Regression models should have been used. Correlations and mean differences does not answer to the research question.
- Confounding factors should be considered
- Post-hoc Bonferroni corrections should have been used.
- Sample size computation and power analysis is not presented.
- Inclusion and Exclusion criteria of participants are missing.
- No information is provided about the therapeutic intervention during the hospital stay of the patients.
- Methods are confusing, Authors should follow the STROBE guidelines
- Conclusions overestimates the results.
Consider that the topic was studied earlier:
-Ones K, Yilmaz E, Cetinkaya B, Caglar N. Quality of life for patients poststroke and the factors affecting it. J Stroke Cerebrovasc Dis. 2005 Nov-Dec;14(6):261-6. doi: 10.1016/j.jstrokecerebrovasdis.2005.07.003. PMID: 17904035.
Author Response
Mrs. Jessica Fernández Solana
Department of health sciences
University of Burgos, Paseo Comendadores s/n.
Burgos, 09001, Spain
Tel. (+34) 947499108
Email: jfsolana@ubu.es
16-12-2022
IJERPH. Subject: Submissions Needing Revision
Dear editor.
Thank you very much for inviting us to submit our response to reviewers for our manuscript (ijerph-2010935) entitled: “A cross-sectional study: determining factors of functional independence and quality of life of patients one month after having suffered a stroke.”
We have checked our manuscript according to the Academic Editor, the reviewers’ comments and the Journal requirements. We have also responded to some comments from reviewers point by point).
We would be very grateful if you could consider our manuscript to be published in your journal.
Yours sincerely,
Jessica Fernández Solana, OT, PT
- Response to Reviewer 3:
First of all, we would like to express our sincere gratitude for all comments and suggestions received from the Reviewer 1. This information has certainly enriched the text for its best understanding, thank you very much indeed. We have clarified the reviewer1’s questions. We have introduced the required changes both in our answers to the specific comments and in the final manuscript V2.
Dear Authors,
Thank you for the opportunity to revise your manuscript. While the topic is relevant, there are serious concerns about the rational, study design, statistics and conclusions.
Please find below major issues:
- In a cross-sectional study is not possible to draw casual relations between variables and outcomes. It is a correlation, that does not mean that a variable causes a change in the outcome. A controlled prospective study is needed.
Response: Thank you for your comment. In this cross-sectional study, the ANOVA test was used to compare the means of two or more groups and to identify whether they are significantly different. In this way, we can find out whether belonging to one group or another has a significant influence on the variables of interest.
- How the variables were selected?
Response: Thank you for your comment. The variables used in the study were collected using a form that included both socio-demographic data and a battery of various scales measuring functionality, cognitive impairment and quality of life. These were therefore the variables taken into account in the study.
- Regression models should have been used. Correlations and mean differences does not answer to the research question.
Response: The objective of the research has been changed to the following: “the main objective of this research is to study which factors are related to functional capacity and QoL a month after suffering a stroke, so that these can be considered in early interventions”.
- Post-hoc Bonferroni corrections should have been used.
Response: This information has been added to the manuscript (see lines 223-225 and 268-269).
- Confounding factors should be considered
Response: This information has been added to the manuscript (see lines 390-391).
- Sample size computation and power analysis is not presented.
Response: This information has been added to the manuscript (see lines 96-101).
- Inclusion and Exclusion criteria of participants are missing.
Response: Inclusion and exclusion criteria have been included:
“This cross-sectional study was conducted in Burgos and Córdoba (Spain). The sample consisted of 81 people who had previously suffered a stroke. As inclusion criteria, it is contemplated that the participants are older than 18 years, with a diagnosis of residual hemiparesis due to stroke, those whose movement of the affected upper limbs is classified between stages II and IV of the Brunnstrom Scale [21].
The study population was recruited at discharge from the Neurology Service and Stroke Unit of both hospitals through consecutive sampling.”
- No information is provided about the therapeutic intervention during the hospital stay of the patients.
Response: The hospital intervention was carried out in the second phase, which resulted in a longitudinal study. Therefore, in this manuscript, it is not considered relevant to explain the intervention performed, the aim of which is not to test the effects of the intervention.
- Methods are confusing, Authors should follow the STROBE guidelines
Response: Thank you for your comment. For the presentation of the methods, the guidelines provided by the journal have been followed.
- Conclusions overestimates the results.
Response: The conclusion has been modified to meet the objective.
Consider that the topic was studied earlier:
Ones K, Yilmaz E, Cetinkaya B, Caglar N. Quality of life for patients poststroke and the factors affecting it. J Stroke Cerebrovasc Dis. 2005 Nov-Dec;14(6):261-6. doi: 10.1016/j.jstrokecerebrovasdis.2005.07.003. PMID: 17904035.
Response: Thank you for your recommendation, has been taken into account in the document
We hope we have now answered all your comments and we are looking forward to hearing from you again.
Thank you very much,
Jessica Fernández Solana, OT, PT

Round 2
Reviewer 2 Report
Dear Authors,
thank you for adressing the majority of my comments.
I suggest to perform MANOVA analysis to better understand how different characteristics works.
Warmely
Author Response
Mrs. Jessica Fernández Solana
Department of health sciences
University of Burgos, Paseo Comendadores s/n.
Burgos, 09001, Spain
Tel. (+34) 947499108
Email: jfsolana@ubu.es
29-12-2022
IJERPH. Subject: Submissions Needing Revision
Dear editor.
Thank you very much for inviting us to submit our response to reviewers for our manuscript (ijerph-2010935) entitled: “A cross-sectional study: determining factors of functional in-dependence and quality of life of patients one month after having suffered a stroke.”
We have checked our manuscript according to the Academic Editor, the reviewers’ comments and the Journal requirements. We have also responded to some comments from reviewers point by point).
We would be very grateful if you could consider our manuscript to be published in your journal.
Yours sincerely,
Jessica Fernández Solana, OT, PT
- Response to Reviewer 2:
First of all, we would like to express our sincere gratitude for all comments and suggestions received from the Reviewer 2. This information has certainly enriched the text for its best understanding, thank you very much indeed. We have clarified the reviewer2’s questions. We have introduced the required changes both in our answers to the specific comments and in the final manuscript V2.
Dear Authors,
thank you for adressing the majority of my comments.
I suggest to perform MANOVA analysis to better understand how different characteristics works.
Warmely
Response: Thank you for your comment. Following your suggestion, a MANOVA test was carried out to estimate the differences between the means of the categorical variables, by means of the joint comparison of the dependent variables (FIMFAM and NEWSQOL), but it was finally decided not to include it in the manuscript as the assumption of homogeneity of covariance matrices was not met (F=1.99; p<0.05).
We hope we have now answered all your comments and we are looking forward to hearing from you again.
Thank you very much,
Jessica Fernández Solana, OT, PT

Reviewer 3 Report
All the comments have been addressed
Author Response
Mrs. Jessica Fernández Solana
Department of health sciences
University of Burgos, Paseo Comendadores s/n.
Burgos, 09001, Spain
Tel. (+34) 947499108
Email: jfsolana@ubu.es
29-12-2022
IJERPH. Subject: Submissions Needing Revision
Dear editor.
Thank you very much for inviting us to submit our response to reviewers for our manuscript (ijerph-2010935) entitled: “A cross-sectional study: determining factors of functional independence and quality of life of patients one month after having suffered a stroke.”
We have checked our manuscript according to the Academic Editor, the reviewers’ comments and the Journal requirements. We have also responded to some comments from reviewers point by point).
We would be very grateful if you could consider our manuscript to be published in your journal.
Yours sincerely,
Jessica Fernández Solana, OT, PT
- Response to Reviewer 3:
First of all, we would like to express our sincere gratitude for all comments and suggestions received from the Reviewer 3. This information has certainly enriched the text for its best understanding, thank you very much indeed. We have clarified the reviewer1’s questions. We have introduced the required changes both in our answers to the specific comments and in the final manuscript V2.
All the comments have been addressed
Response: Thank you for your comments and for reviewing our manuscript.
Thank you very much,
Jessica Fernández Solana, OT, PT
